# Advances in the Pathogenesis and Treatment of Systemic Lupus Erythematosus

**DOI:** 10.3390/ijms24076578

**Published:** 2023-03-31

**Authors:** Daniele Accapezzato, Rosalba Caccavale, Maria Pia Paroli, Chiara Gioia, Bich Lien Nguyen, Luca Spadea, Marino Paroli

**Affiliations:** 1Division of Clinical Immunology, Department of Clinical, Anesthesiologic and Cardiovascular Sciences, Sapienza University of Rome, 00185 Rome, Italy; 2Eye Clinic, Department of Sense Organs, Sapienza University of Rome, 00185 Rome, Italy; 3Post Graduate School of Public Health, University of Siena, 53100 Siena, Italy

**Keywords:** systemic lupus erythematosus, B-cells, T-cells, plasmacytoid cells, type-I interferon, cell-based therapy

## Abstract

Systemic lupus erythematosus (SLE) is a genetically predisposed, female-predominant disease, characterized by multiple organ damage, that in its most severe forms can be life-threatening. The pathogenesis of SLE is complex and involves cells of both innate and adaptive immunity. The distinguishing feature of SLE is the production of autoantibodies, with the formation of immune complexes that precipitate at the vascular level, causing organ damage. Although progress in understanding the pathogenesis of SLE has been slower than in other rheumatic diseases, new knowledge has recently led to the development of effective targeted therapies, that hold out hope for personalized therapy. However, the new drugs available to date are still an adjunct to conventional therapy, which is known to be toxic in the short and long term. The purpose of this review is to summarize recent advances in understanding the pathogenesis of the disease and discuss the results obtained from the use of new targeted drugs, with a look at future therapies that may be used in the absence of the current standard of care or may even cure this serious systemic autoimmune disease.

## 1. Introduction

Systemic lupus erythematosus (SLE) is an autoimmune disease, characterized by phases of flare-ups and remission, that can cause severe damage to many organs and tissues. The organs most affected by SLE are the kidneys, nervous system, joints, and skin. The hallmark of SLE is the production of circulating autoantibodies, with the formation of immune complexes that precipitate in vessels, with activation of potent inflammatory responses ultimately responsible for multi-organ damage [1,2]. Fortunately, in recent decades, the pathogenesis of SLE has been greatly elucidated, with the identification of dysregulation of cells of the innate and adaptive systems. It has also been shown that a predominant role is played by type-I interferon (IFN), which is responsible for the hyperactivation of genes coding for pro-inflammatory molecules by target cells, a phenomenon termed type-I IFN signature [3]. Although SLE therapy is still based on nonspecific immunomodulatory and immunosuppressive drugs [4], new treatments directed against specific targets of the immune system have recently been developed, and some have been approved by regulatory agencies [5]. Such new drugs, however, still need to be combined with conventional therapy to achieve acceptable disease control. In this review, we will discuss new advances in our understanding of the mechanisms underlying the pathogenesis of SLE, the therapeutic potential of available targeted drugs, and the future development of novel therapeutic strategies that will hopefully lead to safe personalized therapy, possibly avoiding combination with conventional therapy.

## 2. Epidemiology

In recent decades, the incidence and prevalence of SLE have increased in all countries. This increase in prevalence has been attributed to better diagnosis and the availability of data from international registries. It is currently estimated that the incidence of overall SLE ranges from 0.3 to 23.2 cases/100,000 person-years, depending on the geographical region considered [6]. The highest incidence has been reported in North America, while countries in sub-Saharan Africa, Europe, and Australia have a lower incidence. These differences have been attributed to different genetic predisposition [7], and socioeconomic [8] and environmental factors [9]. Women of reproductive age are more prone to SLE than males, with the incidence ratio between females and males varying between 8:1 and 15:1 [9]. African American populations have the highest SLE incidence and mortality, followed by Hispanic and Asian populations, while Caucasian populations have the lowest rates of the disease [10]. On the other hand, it has been observed that African populations are more susceptible to SLE and are more resistant to systemic treatment with corticosteroids and immunosuppressants [11]. In total, it has been calculated that the risk of mortality in SLE patients is increased about 2.6-fold compared with the general population [12]. Delayed diagnosis, renal involvement, high disease activity index, presence of infections, and major cardiovascular events were found to be the main predictors of mortality [11].

## 3. Diagnosis, Management, and Activity Criteria for SLE

At least one positive antinuclear antibody (ANA) test is required as an entry criterion for SLE, according to the most recently published classification from the European Alliance of Associations for Rheumatology (EULAR)/American College of Rheumatology (ACR), in 2019 [5]. However, the presence of ANAs is not exclusive to SLE. They can be found in healthy individuals and patients with other autoimmune and nonautoimmune diseases [13,14,15]. Moreover, about 30% of patients with a clinical diagnosis of SLE are ANA-negative [16]. The anti-dsDNA antibody is used for the diagnosis of SLE. This antibody is also strongly associated with disease activity [17]. Anti-extractable nuclear antigen (ENA) antibodies are more specific than ANA for the diagnosis of SLE. In particular, the anti-Sm antibody is a marker associated with SLE. Often, it is associated with the presence of anti-U1-ribonucleoprotein (U1-RNP) antibodies, because both bind small nuclear ribonucleoprotein (snRNP). The presence of such high titer antibodies is found in the serum of patients with mixed connective tissue disease [18]. The characteristic antibodies of Sjögren’s syndrome, anti-SSA and anti-SSB, are found in 24–60% of patients with SLE, and are associated with neonatal SLE [19]. Anti-histone antibodies are associated with drug-induced lupus, while anti-ribosomal antibodies are associated with lupus. Antiphospholipid antibodies, which include lupus anticoagulant, anti-cardiolipin antibodies, and anti-β2 glycoprotein 1 antibody, are important indicators of vascular inflammation and thromboembolic risk. Antiphospholipid antibodies can also be observed in primary antiphospholipid syndrome not associated with SLE, or other inflammatory rheumatic conditions. Such antibodies are associated with recurrent early abortions, thrombosis, and intrauterine livedoid vasculitis, as well as neuropsychiatric and cardiovascular complications [20,21]. In addition, the diagnosis of SLE requires at least ten additive points, accumulated from seven clinical and three immunological domains. The new criteria have been found to have a sensitivity of 96.1% and a specificity of 93.4% [22]. Periodic follow-up and evaluation are necessary for the long-term management of patients after the diagnosis of the disease. Given the complex multisystem features of SLE, accurate measurement of the disease activity remains a challenge [23]. The most widely used tool is the SLE Disease Activity Index-2K (SLEDAI-2K). Patients are therefore classified as having SLE, from mild to severe, depending on a score ranging from ≤ 6 to 12 [24]. The 2004 British Isles Lupus Activity Group (BILAG) index, in eight organ systems, has been found to provide a more comprehensive systems-based measure than SLEDAI-2K [25]. For organ damage assessment, the Systemic Lupus International Collaborating Clinics (SLICC)/ACR damage index (SDI) is an internationally recognized tool [26]. Another disease assessment index commonly used in clinical trials, is the SLE Responder Index (SRI), which integrates criteria from the Safety of Estrogens in Lupus Erythematosus National Assessment (SELENA)-SLEDAI, Physician Global Assessment (PGA), and BILAG 2004 [27]. Figure 1 summarizes the latest classification criteria for the diagnosis of SLE and how disease activity and organ damage are measured.

## 4. SLE Pathogenesis

### 4.1. The Role of Adaptive Immunity

#### 4.1.1. B Cells and Autoantibodies in SLE

B lymphocytes are characterized by the expression on their membrane of the B-cell receptor (BCR). This receptor is physiologically devoted to the recognition of pathogens and the subsequent production of specific antibodies [28]. During the process of B-cell development, autoreactive B cells can also be generated. Although the development of these host-dangerous cells is controlled by immunological tolerance systems, such as clonal deletion or induction of peripheral anergy, these control mechanisms can sometimes fail. This allows the unwanted expansion and activation of such autoreactive B cells, with the possible onset of autoimmune diseases [29,30,31]. After development, B cells, including self-reactive B cells, require the intervention of soluble factors to ensure their survival and proliferation. Among these, the most important is the B-cell activating factor (BAFF), also known as B lymphocyte stimulator (BLys) [32,33]. The repertoire of autoantibodies produced by autoreactive B cells, targets mainly nuclear antigens. A key role in the generation of these autoantibodies is played by toll-like receptors (TLRs). Abnormal engagement of TLRs TLR7 and TLR9 subtypes in SLE, has been shown to effectively promote the production of autoantibodies against double-stranded DNA (dsDNA) and RNA-associated autoantigens, respectively [34,35,36,37]. Long-lived plasma cells (LLPCs) derived from the terminal differentiation of B cells, are an important source of autoantibody production in SLE. Short-lived plasmablasts, after interaction with CD4+ T cells in the germinal centers of the lymph nodes, have been shown to transform into high-affinity plasma cells that migrate to specific niches in the bone marrow, where they are protected from external events, being able to survive for a long time and continuing to produce autoantibodies [38]. Spontaneous formation of germinal centers, favoring the generation of LLPCs, is observed in both murine and human SLE, suggesting that this phenomenon is strictly involved in the genesis of autoantibody production [39]. Importantly, B lymphocytes may also play a role as antigen-presenting cells (APC) to autoreactive T lymphocytes in SLE, as demonstrated in mouse models [40,41]. A debated issue is the pathogenetic role of autoantibodies. Because the presence of autoantibodies can be detected in serum even years before the clinical signs of SLE, this has been considered an indication that these antibodies are a biomarker of the disease rather than a pathogenic factor. However, much evidence suggests their central role in the immunopathogenesis of SLE. Of particular importance, is the observation of the presence of immune complexes in lupus nephritis, at the glomerular level, formed by various autoantibodies, including anti-dsDNA antibodies, whose removal leads to amelioration of the disease [42,43,44]. Moreover, neonatal lupus erythematosus (NLE) develops as a result of the passive transfer of maternal autoantibodies across the placenta, which does not allow the passage of cells, including those of the immune system [45]. From these and other observations, it is possible to conclude that autoantibodies may contribute, at least to some extent, to the clinical manifestations of SLE.

#### 4.1.2. T Cells in SLE Pathogenesis

Self-reactive T cells play a key role in the genesis of SLE. T-helper 1 (Th1) cells play a central role in the pathogenesis of SLE, by promoting oxidative stress related to IFNγ production [46]. In contrast, the number of IL-4-producing Th2 cells is reduced in the peripheral blood of SLE patients, suggesting their potential protective role, and that SLE activity may be associated with an increased IFNγ/IL-4 ratio [47]. T-helper 17 (Th17) cells are also involved in SLE pathogenesis. These cells are the main source of IL-17, a family of cytokines with potent inflammatory effects. In addition to their defensive action against pathogens, members of the IL-17 family can exacerbate tissue injury, because of their pro-inflammatory activity. IL-17 induces neutrophil recruitment, activation of the innate immune system, and enhancement of B-lymphocyte functions [48]. It has been reported that IL-17 levels correlate with SLEDAI in patients with LN [49,50]. Regulatory T cells (Tregs) are critical in maintaining peripheral tolerance to self-antigens. Although quantitative and qualitative differences in Tregs have been described in SLE, studies to date have shown conflicting results, and their role in SLE is still undefined. However, some studies have proposed that these cells, due to their ability to suppress effector T lymphocytes, could be considered in the basic cell therapy of SLE [51,52,53]. T-follicular helper (Tfh) cells are located in germinal centers and extrafollicular foci. These cells have been involved in the generation of autoreactive B-cell clones in murine and human SLE [54]. Tfh cells were found to aggregate in renal tissue with B cells, similar to what is observed in germinal centers in LN [55]. All these findings support the concept that interactions between CD4+ T cells and B cells are crucial in the development of autoimmunity, as they contribute decisively to the development and maintenance of autoreactive B cells and their differentiation into autoantibody-producing plasma cells. CD8+ T cells are also involved in the immunopathogenesis of SLE. Circulating CD8 T lymphocytes of SLE patients exhibit functional defects, including impaired cytolytic function, with reduced production of granzyme and perforin [56]. A depleted phenotype of circulating CD8 T lymphocytes in SLE patients has been associated with lower disease flare rates [57]. However, the qualitative abnormalities of CD8+ T lymphocytes are also related to the susceptibility of SLE patients to infections, which may be further exacerbated by the use of immunosuppressive drugs [58]. Finally, γδ-T lymphocytes were found in a higher percentage in SLE patients than in controls, suggesting their role in the autoimmune response [59,60]. 

### 4.2. The Role of Innate Immunity

#### 4.2.1. Role of Neutrophils in SLE

It has been observed that in SLE, the neutrophil function is abnormal at several levels. First, neutrophils show reduced phagocytic capacity [61] and the inability to remove apoptotic cells, which are a known source of normally hidden self-antigens [62,63]. Variants of *ITGAM*, *NCF1*, and *NCF2* genes, have been reported to be risk factors for SLE development, since they induce alteration of phagocytosis and dysregulate reactive oxide species production (ROS) [64,65,66]. It has also been reported that neutrophils can produce type-I IFN independently of TLRs stimulation and promote abnormal B-cell development in the bone marrow of SLE patients [67,68]. A subtype of neutrophils, called low-density granulocytes (LDG), is highly represented in the peripheral blood of SLE patients. These cells are associated with the presence of IFN signature and disease severity [69,70,71], and activate CD4+ T-cells to produce IFNγ and TNFα [69]. In SLE, LDGs are characterized by an increased ability to form neutrophil extracellular traps (NETs), released during their apoptosis (NETosis) [72]. NETs are rich in decondensed nucleic acids, and chromatin expelled outside the cells during the formation process can induce specific autoreactive immune responses against nucleic acid antigens [73]. Neutrophils are also characterized by the generation of ROS which, in normal conditions are responsible for cell killing, but in SLE contribute to endothelial damage [73]. Several genetic polymorphisms related to neutrophil dysregulation, that increase NET formation, have been described in SLE [74,75,76]. Moreover, neutrophils from SLE patients with mutations resulting in loss of STAT3 function, form NETs more spontaneously than healthy controls [77]. The increased formation of NETs and their reduced clearance may also lead to increased inflammasome activation in macrophages, amplifying the inflammatory response [78]. Taken together, these observations indicate that neutrophils, through NET formation, have very important immunostimulatory effects in SLE, contributing significantly to the immune dysregulation that leads to tissue damage.

#### 4.2.2. Role of Plasmacytoid Dendritic Cells

Lymphoid-origin plasmacytoid dendritic cells (pDCs) are characterized by the ability to produce high levels of type-I IFN, thus playing a key role in the pathogenesis of SLE [79,80,81]. Production of type-I IFN occurs primarily in response to circulating ssRNA and dsDNA, that are internalized by pDCs through Fc-gamma receptor IIA (FcγRIIa). Nucleic acids are then recognized by TLR7 and TLR9 in the cytoplasm [82]. These receptors can also be activated by endogenous nucleic acids present in NETs. Once activated, TLRs trigger signaling pathways mainly involving the *myeloid differentiation response gene-88* (*Myd88*) and interleukin-1 receptor-associated kinase 4 (IRAK4), leading to the activation of interferon regulatory factors 3 (IRF3) and IRF7 for IFN-I production [83,84,85]. The pDC-dependent production of type-I IFN is also important in linking innate and adaptive immunity. This occurs through complex interactions involving monocytes, neutrophils, natural killer cells, and T and B cells [86,87]. In this regard, the production of IFN-I by pDCs, can promote the differentiation of extrafollicular B cells into short-lived plasmablasts, which produce anti-dsDNA antibodies, thus creating a positive feedback loop supporting the autoimmune response, as demonstrated in animal models [88]. Activation of pDCs and high levels of IFN-I production, also increase the number and recruitment of pro-inflammatory T cells. This occurs particularly in the arterial wall. This finding has been associated with the development of accelerated atherosclerosis, as commonly observed in the course of SLE [89]. It should be noted, however, that pDCs may have also a tolerogenic function, through the generation of regulatory T cells (Treg). As mentioned earlier in this review, these cells are known to inhibit the activation of effector T cells [90,91,92]. Moreover, pDCs also facilitate the differentiation of immature B cells into IL-10-producing regulatory B cells (Breg), which can limit IFN-I production by pDCs, thus forming a regulatory feedback, the dysregulation of which is one of the most important components of SLE pathogenesis [93]. PDCs are therefore another possible target of SLE cell therapies. Figure 2 shows the relationships between the innate and adaptive immune systems in the pathogenesis of SLE.

### 4.3. The Role of Mitochondria

Mitochondria are organelles that provide the energy necessary for cell metabolism and survival, being the major source of adenosine triphosphate (ATP) synthesis [94]. These ancestral structures can release high quantities of mitochondrial DNA (mtDNA) after destruction, following cell apoptosis. MtDNA is extremely unstable, and its easy degradation can generate antigenic fragments [95]. MtDNA has been found to induce specific autoreactive T lymphocytes in patients with SLE [96,97]. These in turn may induce B cells to produce anti-DNA antibodies. It has also been observed that sequences of mtDNA are analogous to those of bacteria and therefore able to activate TLRs. TLR recognition triggers powerful downstream inflammatory responses, including type-I IFN production [98]. This pro-inflammatory response, further contributes to the breakdown of tolerance. It has also been reported that several mitochondrial gene variants are linked to the risk of developing SLE, as demonstrated in mouse models [99,100,101]. Mitochondrial polymorphisms increase oxidative stress [102], as demonstrated by the accumulation of oxidized nucleic acids in the mitochondria of neutrophils of patients with SLE [103]. The accumulated oxidized mtDNA can be therefore extruded during NETosis, potentially triggering type-I IFN activation by plasmacytoid dendritic cells (pDCs) [63]. Finally, mitochondrial RNA (mtRNA) is another source of autoantigens, and titers of RNA autoantibodies against mtRNA are significantly higher in patients with SLE compared with controls [104].

### 4.4. The Role of Apoptosis

It has recently been shown that enzymes such as nucleases are critical for nucleic acid digestion and maintenance of tolerance [105]. A deficiency of nucleases is responsible for lupus-like manifestations in mouse models [106]. Recent studies have identified neutralizing antibodies against DNASE1L3 in some SLE patients, resulting in the accumulation of extracellular DNA and the formation of immune complexes [107]. Mice KO for *DNaseII* genes does not survive due to undigested DNA in phagocytes [108,109]. Mice KO for genes encoding TREX1 nuclease develops lupus-like symptoms, including skin lesions, vasculitis, nephritis, and sometimes systemic inflammation [110,111]. These observations lead to the conclusion that undigested nucleosomes are an initial inducer of the autoimmune responses, as observed in SLE patients. Apoptotic cell clearance can induce the breakdown of immune tolerance through several mechanisms. These include the activation of signals mediated by pattern recognition receptors (PRRS) [112,113,114]. The role of apoptosis defects in SLE has been confirmed in human studies, showing that patients with SLE have a defect in apoptotic cell clearance [115,116]. However, it should be noted that, in some mouse models with impaired apoptosis, autoimmunity does not occur, indicating that other events are required to induce the onset of SLE [117]. NETs are also involved in the apoptosis deficit in SLE. The molecular components of NETs can be complexed with DNA, making it resistant to enzymatic digestion by DNases, inducing type-I IFN production by plasmacytoid dendritic cells [118]. 

### 4.5. The Role of Interferons in SLE 

A recent finding on the pathogenesis of SLE that opened new lines of research for innovative drug development, was the recognition of a high type-I IFN signature in SLE patients [119]. IFNs play a pivotal role in defense against pathogens. Three types of IFNs can be distinguished: type-I IFN, which is a family composed of IFNα (13 subtypes), IFNβ, IFNω, IFNκ, and IFNε; type-II IFN, also known as IFNγ; and type-III IFN (IFNλ) [3,120,121]. Type-I IFN, particularly the IFNα and IFNβ family members, is the one mainly involved in the pathogenesis of SLE. Type-I IFN is induced by the activation of pathogen recognition receptors (PRRs) such as toll-like receptors (TLRs), retinoic acid-inducible gene I (RIG-I), and melanoma differentiation-associated protein 5 (MDA5). All these molecules are activated by nucleic acids or by bacterial products such as lipopolysaccharides and peptidoglycan [122]. Although virtually any cell type can produce type-I IFN [123], very high levels of this cytokine are synthesized by pDCs, as already discussed [124,125,126]. Type-I IFNs recognize the IFNα receptor (IFNAR), a heterodimeric complex that in turn activates intracellular signaling through Janus kinase 1 (JAK1) and tyrosine kinase 2 (TYK2). These proteins phosphorylate transcriptional signal transducers and activators STAT 1 and STAT 2. These intracellular molecules bind IRF7 and IRF9, to form the ISGF3 complex. This complex translocates into the nucleus, where it induces transcription of genes named *IFN-sensitive response elements* (*ISREs*), encoding for proteins contributing to the inflammatory cascade [127,128]. Initial experimental animal model studies showed that the administration of type-I IFN was able to induce the production of autoantibodies and contribute to organ damage [129]. The earliest evidence suggesting that type one interferon could play a key role in the genesis of SLE in humans, came from the observation that patients treated with IFNα for hepatitis C [130] or neoplastic disease [131,132], could develop antinuclear antibody positivity and in some cases lupus-like syndromes. These clinical conditions regressed after discontinuation of IFNα treatment [133,134]. It has been shown that polymorphisms in genes along the type-I IFN signaling pathways represent important genetic risk factors for the occurrence of SLE-including *IRF* genes [135,136,137]. Polymorphisms have been also described in *STAT4*, *STAT3*, and *TYK2* genes [77,138]. The IFN signature is also emerging as a possible biomarker for precision treatment with novel anti-IFN therapeutic agents, as discussed in more detail in the section on SLE therapy. 

## 5. SLE Treatment

### 5.1. The EULAR/ACR Recommendations

Unlike other rheumatic diseases, the treatment of SLE has so far not taken decisive steps to replace traditional therapy [139]. According to recent recommendations of the European League Against Rheumatism (EULAR), the goal of treatment is to achieve remission, or at least a state of low disease activity [4,140,141]. The concept of "treat-to-target," initially formulated for the treatment of rheumatoid arthritis, has thus been extended to SLE therapies as well [4]. EULAR and the ACR periodically update their recommendations for the treatment of SLE. Regarding pharmacological treatment with conventional disease-modifying drugs (csDMARDs), the use of hydroxychloroquine (HCQ) was recommended for all patients, at a dose of no more than 5 mg/kg body weight, to minimize retinal complications. HQC, a drug that has been shown to be safe in the long term, reduces episodes of disease flare and the occurrence of cardiovascular events [142,143,144]. It also has a low cost, that makes it usable even in low-income geographic areas. Therefore, it is likely that this drug will be used for a long time to come in the treatment of SLE, in combination with innovative drugs. For chronic treatment, glucocorticoids should be reduced to less than 7.5 mg/d prednisone equivalent and, if possible, discontinued. In fact, steroids are burdened with side effects when administered for prolonged periods or at high doses, such as induction of osteoporosis, worsening of diabetes and hypertension, and susceptibility to infections [145]. Interruption of steroid administration is sometimes possible, through the use of so-called steroid-sparing agents, that include the immunosuppressants azathioprine, methotrexate, mycophenolate mofetil, and cyclophosphamide (CYC). However, side effects and toxicity limit the use of these drugs. Mycophenolate, generally more effective than azathioprine, is teratogenic and cannot be used during pregnancy [146]. CYC has been shown to be highly effective in organ-threatening manifestations, such as central nervous system involvement and LN, even at a lower dose than before [147]. CYC, however, is burdened by several important side effects, including cancer [148]. Therefore, for patients who do not respond, or respond poorly, to conventional immunosuppressants, to avoid severe side effects and allow safer pregnancy, new therapies are highly desirable. As pointed out earlier, the reduction in steroids is a particularly important recommendation, although they remain an important and often unavoidable therapeutic adjunct. In addition, the use of other conventional immunosuppressive therapies is still necessary. For all these considerations, additional treatment with the available innovative or biologic drugs is recommended only in addition to conventional therapy, when the latter is not sufficient to control the disease [149].

### 5.2. Rituximab

The use of rituximab, a chimeric anti-CD20 monoclonal antibody capable of causing B-lymphocyte depletion, has a great rationale for use in SLE therapy. However, two major clinical trials failed to achieve their primary endpoints, probably due to faulty study design. Notably, both clinical trials, EXPLORER for moderately-to-severely active SLE [150], and LUNAR for lupus nephritis [151], failed to meet their respective endpoints. However, it should be noted that some important favorable therapeutic effects of rituximab therapy have emerged from further data analysis. In this regard, patients treated with rituximab showed improvements in proteinuria in the LUNAR study [151], and a subgroup analysis of African Americans and Hispanics, suggested a clinical benefit in the EXPLORER study [150]. Another possibility, besides a flawed study design, that may explain the disappointing results obtained with the use of rituximab in SLE, is the inefficient depletion of tissue-resident CD201 B cells. Indeed, although circulating B cells are rapidly depleted, the persistence of B cells within inflamed target organs has been observed in patients with rheumatoid arthritis or Sjogren’s syndrome treated with rituximab [152,153]. In mouse models of lupus, complete ablation of B cells by treatment with chimeric anti-CD19 receptors, has been shown to produce lasting remission of the disease [154]. Thus, only complete ablation of peripheral B cells may correlate with an effective therapeutic response. This was highlighted in a post hoc analysis of data obtained from the LUNAR study [155]. For this reason, the use of novel anti-CD20 agents, designed for more efficient B-cell ablation, has been proposed [156]. Obinutuzumab, a type II monoclonal antibody showing more effective B-cell depletion than rituximab [156,157], is currently being tested in class III/IV lupus nephritis (NOBILITY study). The available data show that patients with active class III/IV nephritic lupus treated with obinutuzumab, have improved renal response, with 12.7% greater efficacy than a placebo. However, some infectious complications have emerged during therapy with this agent [158]. 

### 5.3. Belimumab

Anti-BAFF belimumab received Food and Drug Administration (FDA) approval in 2011, for the treatment of moderate to severe SLE, in patients older than 18 years. This approval was based on the efficacy results of the BLISS-52 and BLISS-76 trials [159,160]. In BLISS-52, enrolled subjects were randomized to receive standard therapy plus belimumab, administered intravenously, or placebo, every 4 weeks, with a loading dose administered at week 2. After 52 weeks, subjects receiving belimumab 10 mg/kg or 1 mg/kg, showed a statistically higher probability of achieving an SRI-4 response than those receiving placebo (57.6% and 51.4% vs. 43.6%, respectively). Similarly, in BLISS-72, subjects randomized to belimumab administered intravenously at the same dosages as BLISS-52, achieved an SRI-4 response in a percentage statistically more significant than those who received placebo (43.2% and 40.6% vs. 33.5%, respectively). It was also observed that, the patients who were most likely to respond to therapy, were those with high disease activity (SELENA-SLEDAI ≥ 10), hypocomplementemia, and anti-dsDNA positivity and/or prednisone use at baseline [161]. A reduction in organ damage was also demonstrated in subsequent studies [162,163,164]. The subcutaneous route of administration of belimumab was evaluated in the BLISS-SC study, in which treated subjects achieved an SRI-4 in 61% of cases, compared to 48% of those who received the placebo [165]. These results led to the approval of subcutaneous belimumab, in 2017, for active SLE. In 2019, belimumab was approved for use in pediatric patients with SLE, following an international multicenter study (PLUTO) in subjects aged 5 to 17 years. In this study, subjects who received belimumab achieved a significantly higher percentage of SRI-4 than subjects who received the placebo (52.8% vs. 43.6%) [166]. Belimumab, both intravenous and subcutaneous, then received FDA approval also for the treatment of active lupus nephritis, based on the results obtained in the BLISS-LN study [167]. In this phase III study, patients with class III, IV (with or without class V), or V LN were treated intravenously with belimumab or placebo, along with standard therapy. The patients who received belimumab were significantly more likely to achieve a primary efficacy of renal response (PERR) at week 104. PERR was defined by a urinary protein/creatinine ratio (UPCR) ≤ 0.7, an estimated glomerular filtration rate (eGFR) not less than 20% of basal value, or ≥60 mL per minute. A complete renal response (CRR) was defined as a urinary protein ratio < 0.5, an eGFR not ≥10% less than the basal value, or ≥90 mL per minute. The results showed that, patients treated with belimumab achieved a 43% response compared with 32% in the placebo group, and 30% compared with 20% in the placebo group, in PERR and CRR, respectively. The effect of belimumab in the preservation of kidney function was further confirmed in secondary data analysis [168].

### 5.4. Anifrolumab

Anifrolumab is a fully human immunoglobulin G1 monoclonal antibody, that targets and inhibits type-I IFN receptor subunit 1 signaling. Anifrolumab was approved for the treatment of moderate to severe SLE in combination with standard of care, at a dose of 300 mg administered every four weeks intravenously. Approval by the regulatory agencies was based on efficacy data from the phase IIb MUSE study and the phase III TULIP-1 and -2 studies [169]. In the MUSE trial, a group of patients was randomized to receive anifrolumab 300 or 1000 mg intravenously, or placebo, every 4 weeks for 48 weeks. Patients also received glucocorticoids (GC), an antimalarial, azathioprine, mizoribine, mycophenolate mofetil or mycophenolic acid, or methotrexate [170]. The primary endpoint was the percentage of patients achieving the Systemic erythematosus lupus response index-4 (SRI-4) at week 24, with concomitant oral GC dose reduction at <4 weeks [170]. More patients treated with anifrolumab 300 mg (34.3%) and 1000 mg (28.8%), responded significantly to treatment compared to placebo (17.6%). Better responses were also observed for systemic lupus erythematosus responder index-4 (SRI-4) and BILAG-based composite lupus assessment (BICLA) at 52 weeks. Therefore, two phase III studies, Treatment of Uncontrolled Lupus via the Interferon Pathway (TULIP)-1 and TULIP-2, were conducted. TULIP-1 included an additional lower dose of anifrolumab of 150 mg. The placebo-controlled TULIP-1 study, involved the administration of anifrolumab every 4 weeks for 48 weeks [171]. The primary endpoint was the achievement of SRI-4 at week 52. There was no significant difference in the percentages of patients who reached the primary endpoint between the anifrolumab 300 mg group and the placebo group (36% vs. 40% of patients, respectively), out of a total of 364 subjects who completed the study. In addition, no significant response in the anifrolumab 150 mg group was observed. However, post hoc analysis considering the results, taking into account NSAID use, showed that all outcome measures improved, even though the primary endpoint had not yet been achieved. Of note, the BICLA responses at week 52 were in favor of anifrolumab 300 mg (46%) compared with placebo (30%). Pharmacodynamic evaluation of patients with a type-I IFN-high genetic signature, showed a neutralization of the type-I IFN 21 gene panel of 12.6% as early as week 12 and throughout the study period, for the anifrolumab 300 mg group, but not for the placebo group. Therefore, the experience of the TULIP-1 study led to a modification of the study protocol of the sister study, TULIP-2, before data unblinding [172]. Specifically, the primary endpoint was changed from an SRI-4 response to a BICLA response, at week 52. Disease relapse was defined as ≥1 new BILAG-2004 A-item or ≥2 new BILAG-2004 B-items, as compared with the baseline. After this protocol modification, TULIP-2 met the primary endpoint, as 47.8% of patients on anifrolumab, compared with 31.5% in the placebo group, responded according to the above BICLA criteria. Secondary outcomes were also achieved in the type-I IFN-high group (48% vs. 30.5% in the placebo group), with a reduction in GC use and ≥50% of cutaneous LE activity index (CLASI) at week 12 (49% vs. 25%). Data from the two TULIP studies were then pooled and analyzed [173,174]. More patients in the anifrolumab 300 mg group achieved a BICLA response than those in the placebo group (47.5%). Similarly, more patients in the anifrolumab 300 mg group achieved a response to SRI-4 (52.2% vs. 40.1%), sustained tapering of GC (50.5% vs. 31.8%), a ≥50% reduction in CLASI activity (46% vs. 24.9%), and ≥50% reduction in active joints (49.4% vs. 36.8%), compared with the placebo-treated group [173]. Moreover, the group treated with anifrolumab 300 mg, had a lower annualized flare rate (AFR) than the placebo (0.51% vs. 0.8%), a longer median time to first relapse (140 days vs. 119 days), and fewer patients with ≥1 relapse (33.6% vs. 42.9%). Importantly, among patients who achieved sustained reductions in GC, more patients remained flare-free with anifrolumab (40%) compared with placebo (17%). In the TULIP-LN study, the primary endpoint was the change in the 24-hour urinary protein–creatinine ratio (UPCR) at week 52, for the combined anifrolumab (BR and IR) groups compared with placebo [175]. The secondary endpoint was a CRR, defined as 24-hour UPCR ≤ 0.7 mg/mg, eGFR ≥ 60 ml/min/1.73 m^2^, or no decrease ≥ 20% from baseline. There was no difference in the primary endpoint between the anifrolumab-treated group and the placebo group. However, there were numerical improvements in the CRR achieved by the IR group compared with placebo (45.5% vs. 31.1%), and in the sustained reduction in GC dose (55.6% vs. 33.3%) in the same group. Overall, these data confirmed the efficacy of anifrolumab on many clinical outcomes, including BICLA and SRI-4, mucocutaneous and musculoskeletal manifestations, lower flare rates, relapse rates, and successful tapering from GC to ≤7.5 mg/day. All of these data led, as mentioned, to the approval of anifrolumab as an adjunctive treatment of active moderate to severe SLE, at the recommended dose of 300 mg, administered as a 30-minute intravenous infusion every four weeks, with possible discontinuation of treatment if no improvement in disease control is observed after six months. Table 1 summarizes the main clinical trials of new drugs currently used in SLE therapy.

### 5.5. Voclosporin

Voclosporin (VCS) is an oral calcineurin inhibitor, belonging to the same drug class as tacrolimus and cyclosporine. VCS was approved in January 2021, by the FDA, for the treatment of active lupus nephritis, in combination with conventional immunosuppressive therapy [176]. Two randomized controlled trial (RCT) studies demonstrated improved renal response rate and reduced proteinuria when VCS was added to mycophenolate mofetil (MMF) and steroids, compared with the group receiving MMF and steroids alone [177,178]. Preliminary interim data from a two year extension study, showed sustained reductions in proteinuria and stability of renal function after VCS therapy for up to 30 months [70]. 

## 6. Future Therapies

### 6.1. Targeting of Plasma Cells

LLPCs, which are responsible for antibody production in the bone marrow, do not express CD19 and CD20 molecules and do not require BAFF for long-term survival [179]. Therefore, they are resistant to available B-cell-targeted therapies. This observation suggested that plasma cell depletion might be a more effective target in SLE therapy. To this end, several strategies, such as inhibition of the proteasome, which is necessary for LLPCs’ survival, are being investigated. In animal models, proteasome inhibitor bortezomib reduced the number of plasma cells and prolonged survival in a mouse model of SLE [180]. However, in a small randomized trial in human SLE, a high rate of treatment discontinuation was found, due to severe side effects and minimal impact on dsDNA titers [181]. Possibly, compensatory augmentation of constitutive proteasome components may protect plasma cells from death, by significantly limiting therapeutic capping [182]. Other studies have exploited the experiences of onco-hematology in the treatment of multiple myeloma [183]. In this regard, pilot studies have shown that daratumumab, an anti-CD38 monoclonal antibody expressed on plasma cells, was able to induce significant clinical benefits in refractory SLE [184]. However, because CD38 is expressed by other cells of the immune system important for pathogen defense and immunoregulation, alternative strategies to target plasma cells in SLE should be pursued. In this regard, potential therapeutic agents appear to be elotuzumab, targeting SLAMF7, which is expressed in myeloma cells and nonmalignant plasma cells, as well as activated DN2 cells and circulating antibody-secreting cells (ASCs), in human SLE [185]. 

### 6.2. Cell-Based Therapies

#### 6.2.1. Hematopoietic Stem Cell Transplantation

The hematopoietic stem cell transplantation (HSCT) approach, commonly used in hematology, was initially employed for life-threatening SLE. In recent times, thanks to improvements in biotechnology, this treatment option has become a possibility for patients who are less severe but refractory to standard therapy [186,187]. The rationale for this therapy is to eliminate self-reactive memory T and B lymphocytes from the recipient’s immune system, as well as plasma cells that, as described above, are refractory to standard B-cell depletion therapy and are not sensitive to anti-BAFF agents, and to replace them with normal cells. In contrast, plasma cells are susceptible to conditioning treatment in the presence of anti-thymocyte globulin, to avoid graft-versus-host disease (GVHD), followed by regeneration of the hematopoietic and immune system by stem cell transplantation [188,189,190]. To date, more than 300 patients have received autologous transplantation for SLE. Reported results have shown that 50%–66% of treated patients achieved remission at five years after discontinuation of immunosuppressive therapy [191]. Most recent studies have reported treatment-related mortality to be less than 5%, falling significantly from the initial studies. Patients who respond, are usually free of clinical symptoms, and can regain seronegativity for antinuclear antibodies, a condition very difficult to achieve with standard therapy [140]. Early use of HSCT has also been found to protect against organ failure and drug toxicity, and to improve quality of life [192]. Allogeneic HSCT (allo-HSCT) can also be used to restore a dysfunctional immune system, although its wide application has been limited by the risk of graft-versus-host disease (GvHD) and other procedure-related complications. A retrospective analysis of the EBMT registry, published in 2019, reported five SLE patients successfully treated with allo-HSCT, and three other SLE cases in the literature achieved complete remission of autoimmune manifestations [193,194,195,196]. This evidence provides the principle for the use of allo-HSCT as a potentially curative approach.

#### 6.2.2. Chimeric Antigen Receptor T Cells

Cancer immunotherapy has experienced a new phase of development in recent years. One method is to artificially harness the immune system to fight cancer cells, after the fusion of T cells with chimeric molecules specific for tumor antigens. Such genetically engineered cells are called chimeric antigen receptor T cells (CAR-T). Chimeric antigen receptors (CARs) are composed of an extracellular domain and an intracellular portion. The extracellular domain is responsible for target recognition, being in most cases derived from the light chain and heavy chain of the variable portion of monoclonal antibodies bound together [197,198,199]. The intracellular portion mediates signal transduction in the T cell, and consists of one or more domains [200]. These two portions are linked together by a peptide linker or spacer [201,202]. CARs have been extensively studied in the treatment of cancer. Currently, several products, based on autologous CAR-T cells targeting the surface antigen CD19, have been approved for the treatment of B-cell malignancies [203]. Recently, the first data were published on the use of an anti-CD19 CAR-T cell-based strategy in a patient with refractory SLE. These data demonstrated rapid clinical remission, with no notable adverse effects, accompanied by sustained depletion of circulating B lymphocytes and rapid disappearance of native serum anti-DNA antibodies [204]. Subsequently, four additional patients with SLE, presenting with a form of disease refractory to standard therapy, were treated. Preliminary results on safety and efficacy were encouraging, but data on long-term follow-up are needed [205]. The toxicities of the therapy consist mainly of cytokine storm syndrome, that can be life-threatening for patients, although in most cases effective specific strategies can be used that can block released pro-inflammatory cytokines, including the use of monoclonal antibodies that can block IL-6 activity [206]. In SLE, B-cell depletion is a potentially curative approach, although a complete blockade of antibody production may increase susceptibility to infection. For these considerations, CAR-T cells represent an interesting and promising approach for SLE therapy, but some questions remain open, such as the duration of responses during B-cell repopulation and the identification of an appropriate target population.

### 6.3. Other Promising Therapies

Some attempts have been made to enhance Treg in SLE, due to their immunosuppressant properties, with promising results, by adoptive transfer of polyclonal Treg cells [207,208]. The possibility of targeting pDCs with BDCA2-specific antibody, because of the critical role of such cells in the pathogenesis of SLE, is an attractive therapeutic strategy [209]. The administration of allogeneic mesenchymal cells with immunoregulatory properties represents a further therapeutic opportunity [210]. In preclinical studies, inhibition of peptidyl arginine deiminase 4 (PAD4), an enzyme playing an important role in NET formation, has been found to exert a favorable effect in nephritis mouse models [211]. Inhibition of various receptor-associated kinases of several pro-inflammatory molecules, including JAK1 and TYK2, has shown promise, and RCT studies are underway to evaluate their efficacy [212]. Other studies are examining strategies to increase the immunosuppressive activity of regulatory T cells, through their activation by low-dose IL-2 administration [213]. Table 2 shows the innovative therapies that are still under investigation for possible future treatment of SLE.

## 7. Conclusions

SLE is a disease whose genesis is not yet well understood. The involvement of both the innate and adaptive immune systems underscore its great complexity. However, recent advances in research have made it possible to unravel at least some of the fundamental mechanisms underlying the disease. The identification of the key role played by type-I IFN has given further impetus to research. The results obtained so far have made it possible to develop new selective therapies, but these have proven effective only in combination with conventional therapy. In addition, other important issues need to be addressed. These include, assessing whether the side effects of the newly listed agents are acceptable, compared with steroids and conventional therapies. In addition, the high price of the new drugs may not be sustainable in the long term in most countries; finally, it has not yet been clarified for how long the new drugs should be administered. Further research is therefore needed to develop safe and effective therapies, that can be used without the need for steroids and conventional immunosuppressive drugs, which are burdened by many side effects, with a significant impact on patient morbidity and mortality

## Figures and Tables

**Figure 1 ijms-24-06578-f001:**
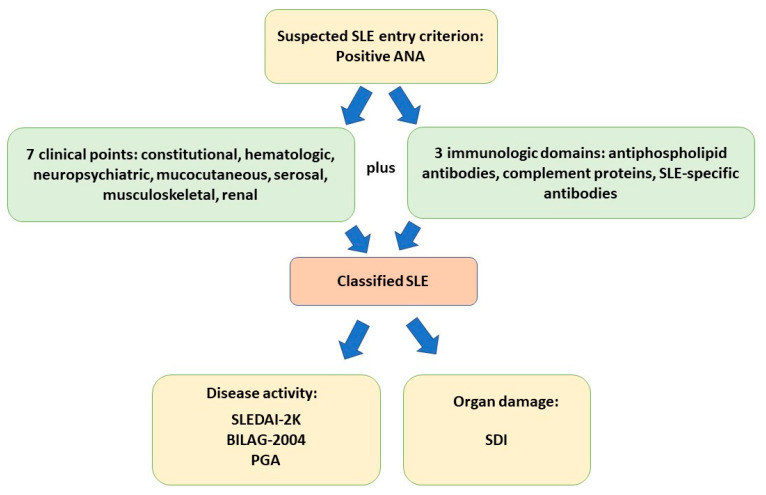
SLE classification according to EULAR/ACR 2019 criteria and measurement of disease activity and organ damage. The entry criterion consists of antinuclear antibody (ANA) positivity on at least one occasion. Definitive diagnosis requires 10 points derived from clinical and immunologic criteria. Disease activity is assessed by the SLE Disease Activity Index-2000 (SLEDAI-2K), British Isles Lupus Activity Group 2004 (BILAG 2004), and Physician Global Assessment (PGA). Organ damage is assessed by the SLICC/ACR damage index (SDI).

**Figure 2 ijms-24-06578-f002:**
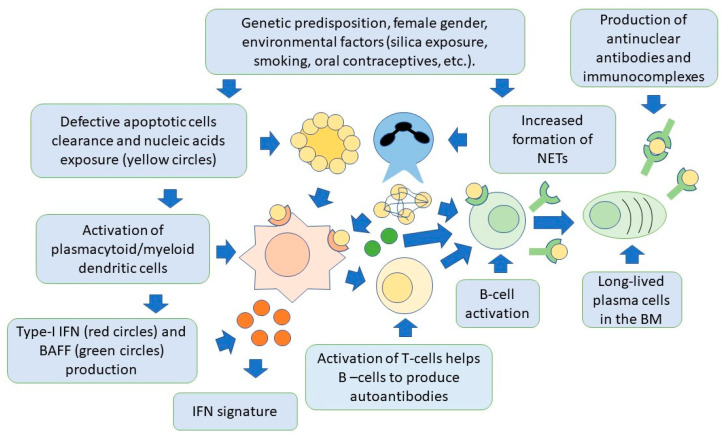
Both innate and adaptive immunity participate in the pathogenesis of SLE. In individuals with a genetic predisposition, and with the contribution of environmental factors, there is an accumulation of apoptotic cells and activation of NET production, by neutrophils. Cell nucleic acids are then exposed to the immune system and activate dendritic cells via toll-like receptors, to produce type-I IFN. This cytokine is responsible for activation of specific genes for pro-inflammatory factors by target cells (IFN signature). Dendritic cells also produce BAFF, which is necessary for B-cell activation and survival, and activate T cells, through the presentation of nuclear self-antigens. Self-reactive B cells are then activated to produce antibodies, and differentiate into long-lived plasma cells, localized in niches in the bone marrow, which are an additional source of autoantibodies. Autoantibodies form immunocomplexes with specific nuclear self-antigens that precipitate in tissues, contributing to organ damage.

**Table 1 ijms-24-06578-t001:** Major clinical trials of new drugs currently used for SLE therapy.

Drug	Target	Molecular Structure	Trial	Dosing	Primary Endpoint (PE)	Result	FDA Approval
Rituximab	Pan-B-cell marker CD20	Chimeric mAb	EXPLORER	1000 mg or placebo on days 1, 15, 168, and 182	BILAG response versus placebo at week 52	PE not met	No
Rituximab	_	_	LUNAR	1000 mg or placebo on days 1, 15, 168, and 182	Complete or partial response at week 52 in LN patients	PE not met	No
Belimumab	BAFF	Human mAb IgG-1 lamba	BLISS-52	10 mg/kg or 1 mg/kg or placebo every 4 weeks	SRI-4 response versus placebo at week 52	PE met	_
Belimumab	_	_	BLISS-76	10 mg/kg or 1 mg/kg i.v. or placebo every 4 weeks	SRI-4 response versus placebo at week 52	PE met	Yes (adults with ANA+, active SLE plus standard therapy)
Belimumab	_	_	BLISS-SC	200 mg s.c. weekly	SRI-4 response versus placebo at week 52	PE met	Yes (adults with ANA+, active SLE plus standard therapy)
Belimumab	_	_	BLISS-LN	10 mg/kg i.v. or placebo every 4 weeks	PERR at week 104 in patients with active LN	PE met	Yes (adults with active LN plus standard therapy)
Belimumab	_	_	PLUTO	10 mg/kg i.v. or placebo every 4 weeks	SRI-4 response versus placebo at week 52 in children aged 5 to 17 years	PE met	Yes (children 5 years and older with SLE and LN plus standard therapy)
Anifrolumab	Type-I IFNR subunit 1	Human mAb IgG-1 kappa	TULIP-1	300 mg or 150 mg or placebo every 4 weeks	SRI-4 response of anifrolumab 300 mg versus placebo at week 52	PE not met	_
Anifrolumab	_	_	TULIP-2	300 mg or placebo every 4 weeks	BICLA response of anifrolumab 300 mg versus placebo at week 52	PE met	Yes (adults with moderate to severe SLE plus standard therapy)
Voclosporin	T-cell inhibition and kidney podocytes stabilization	Calcineurin inhibition	AURORA	Voclosporin 23.7 mg twice daily + MM 1 g daily or MM 1 g daily	CRR in voclosporin + MM versus MM alone at week 52	PE met	Yes (adults with active LN plus standard therapy)

mAb = monoclonal antibody; BAFF = B-cell activating factor; IFNR = interferon receptor; PERR = primary efficacy renal response; CRR = complete renal response.

**Table 2 ijms-24-06578-t002:** Promising future therapies for SLE.

Drug	Mechanism of Action	Reference
Ocrelizumab	CD20+ B-cell depletion	[158]
Bortezomib	Proteasome inhibition of LLPCs	[181]
CAR T cells	CD19+ B-cell depletion	[205]
BDCA2	Anti-pDC antibody	[209]
IL-2	Treg enhancement	[208]
JAK inhibitors	Type-I and type-II IFN signaling inhibition	[212]

LLPCs = long-lived plasma cells; pDC = plasmacytoid dendritic cells; Treg = T regulatory cells; JAK = Janus kinase.

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
