# Peer review of "Advances in the Pathogenesis and Treatment of Systemic Lupus Erythematosus"

_ijms, 2023, doi:10.3390/ijms24076578_

Round 1
Reviewer 1 Report
Congratulatons! Vey interesting and well writen paper.
Author Response
Thank you for appreciating our manuscript!
Reviewer 2 Report
Dear authors, an excellent overview and new treatment options.
I think this work is valuable for publishing.
Kind regards,
Dragana Lazarevic
Author Response

(The authors gave the same response as above.)

Reviewer 3 Report
The authors insist that recent advances in research have made it possible to unravel at least some of the fundamental mechanisms underlying SLE. Type-I IFN plays important role in SLE. The results obtained so far have made it possible to develop new selective therapies, but these have proven effective only in combination with conventional therapy. Further research is therefore needed to develop safe and effective therapies shortly that can be used without the need for steroids and conventional immunosuppressive drugs which are burdened by many side effects with a significant impact on patient morbidity and mortality.
This review is well written and has no major issues. I have a few comments
1. Epidemiological considerations such as the long-term prognosis of SLE, major causes of death, prevalence, and racial differences are also necessary.
2. As for the explanation of existing treatment methods in 4.1, please present data such as treatment success rate and prognosis improvement effect, if any. Please show the current problem with data why a new treatment method is necessary.
3. Are the side effects of the new agents listed in table 1 acceptable compared to steroids? Also, is the price of the drug at a level where long-term prescriptions are possible? How long do you need to administer it?
Author Response
- Question: Epidemiological considerations such as the long-term prognosis of SLE, major causes of death, prevalence, and racial differences are also necessary. Answer: A chapter on epidemiology od SLE was added containing information requested by the reviewer. Bibliographic references related to the text have also been added
- Question: As for the explanation of existing treatment methods in 4.1, please present data such as treatment success rate and prognosis improvement effect, if any. Please show the current problem with data why a new treatment method is necessary. Answer: Chapter 4.1 (now 5.1) has been supplemented with the required information and relevant literature references
- Question: Are the side effects of the new agents listed in table 1 acceptable compared to steroids? Also, is the price of the drug at a level where long-term prescriptions are possible? How long do you need to administer it? Answer: A commentary on these important considerations was added to the conclusions.
All changes made to the original manuscript have been highlighted in yellow.
We thank the reviewer for the important comments that allowed us to significantly improve our manuscript.
Reviewer 4 Report
The manuscript provides a comprehensive, insightful and up-to-date review of the main biological and clinical features of SLE. The text is excellently written, although a few typos and less clear points could be corrected.
Minor remarks
- In section 2., fifth sentence, it should be mentioned "African American populations have the highest SLE incidence". It was earlier stated that African countries have the lowest incidence, so the region of the world must be specified for readers to understand the ethnicity and regional implications.
- In the title and parts of section 4.1.1., "B-cells" should be written "B cells"
- In section 6.1., I do not understand what the authors mean with "experiences of onomatology", because onomatology is the science or study of the origin and forms of proper names.
- In the first sentence of section 5.4., it could be written that anifrolumab is an antibody that targets and inhibits type-I IFNR1 signaling.
- The second sentence of section 6.3. is a confusing, perhaps due to the lack of commas. It is not clear if "an attractive strategy" refers to pDC or to specific antibody.
- In section 6.3., activation by low-dose IL-2 administration is mentioned twice. Perhaps the text could be rearranged to avoid repetition.
- Some abbreviations should be spelled out, to facilitate understanding by less experienced readers: FDA (section 5.3.), RCT (section 5.5.), ASCs (section 6.1.)
- I suggest abbreviating voclosporin as VCS, because it is more frequently used and follows the letter order in the word VoCloSporin.
- In section 6.1., it is not necessary to define the LLPCs abbreviation, because it was already abbreviated in section 4.1.1.
- Typos:
- Ref. 66, in section 3., misses a square bracket
- In Table 1, calcioneurin should be calcineurin
- In section 6.2.1., correct "cells plasma cells" and the number 56 after "immunosuppressive therapy"
Author Response
We initially thank the reviewer for helpful comments and general appreciation of our review. Text changes have been highlighted in green
1) QUERY: In section 2., fifth sentence, it should be mentioned "African American populations have the highest SLE incidence". It was earlier stated that African countries have the lowest incidence, so the region of the world must be specified for readers to understand the ethnicity and regional implications. ANSWER: In accordance with the reviewer's request, it was specified that African American populations have the highest incidence of SLE and that African regions with the lowest incidence are those in the sub-Saharan region.
2) QUERY: In the title and parts of section 4.1.1., "B-cells" should be written "B cells". ANSWER: “B-cells” as been changed to “B cells”
3) QUERY: In section 6.1., I do not understand what the authors mean with "experiences of onomatology", because onomatology is the science or study of the origin and forms of proper names. ANSWER: The term "onomatology" has been corrected to "onco-hematology." We appreciate the reviewer's sense of humor in pointing out the out-of-context meaning of the original term that was the result of a typo.
4) QUERY: In the first sentence of section 5.4., it could be written that anifrolumab is an antibody that targets and inhibits type-I IFNR1 signaling. ANSWER: It has been specified that anifrolumab targets IFNR1 type I signaling.
5) QUERY: The second sentence of section 6.3. is a confusing, perhaps due to the lack of commas. It is not clear if "an attractive strategy" refers to pDC or to specific antibody. ANSWER: To improve clarity, the sentence was rewritten as "The possibility of targeting pDCs with BDCA2-specific antibody because of the critical role of such cells in the pathogenesis of SLE is an attractive therapeutic strategy".
6) QUERY: In section 6.3., activation by low-dose IL-2 administration is mentioned twice. Perhaps the text could be rearranged to avoid repetition. ANSWER: The text has been modified in accordance with the reviewer's suggestion to avoid repetition.
7) QUERY: Some abbreviations should be spelled out, to facilitate understanding by less experienced readers: FDA (section 5.3.), RCT (section 5.5.), ASCs (section 6.1.). ANSWER: The abbreviations have been spelled out
8) QUERY: I suggest abbreviating voclosporin as VCS, because it is more frequently used and follows the letter order in the word VoCloSporin. ANSWER: The abbreviation has been correctly changed from VSC to VCS
9) QUERY: In section 6.1., it is not necessary to define the LLPCs abbreviation, because it was already abbreviated in section 4.1.1. ANSWER: The definition of LLPCs has been removed in this section.
10) QUERY: Ref. 66, in section 3., misses a square bracket. ANSWER: Bibliographic reference [66] was permanently deleted because it was incorrect in the indicated text position
11) QUERY: In Table 1, calcioneurin should be calcineurin. ANSWER: The term "calcioneurin" was changed to "calcineurin"
12) QUERY: In section 6.2.1., correct "cells plasma cells" and the number 56 after "immunosuppressive therapy". ANSWER: The term "cell plasma cells" was changed to "plasma cells," and the number 56 was removed.